# Mesophiles vs. Thermophiles: Untangling the Hot Mess of Intrinsically Disordered Proteins and Growth Temperature of Bacteria

**DOI:** 10.3390/ijms25042000

**Published:** 2024-02-07

**Authors:** Alibek Kruglikov, Xuhua Xia

**Affiliations:** 1Department of Biology, University of Ottawa, 30 Marie Curie, Station A, P.O. Box 450, Ottawa, ON K1N 6N5, Canada; 2Ottawa Institute of Systems Biology, Ottawa, ON K1H 8M5, Canada

**Keywords:** intrinsically disordered proteins, protein structure, thermophilic bacteria

## Abstract

The dynamic structures and varying functions of intrinsically disordered proteins (IDPs) have made them fascinating subjects in molecular biology. Investigating IDP abundance in different bacterial species is crucial for understanding adaptive strategies in diverse environments. Notably, thermophilic bacteria have lower IDP abundance than mesophiles, and a negative correlation with optimal growth temperature (OGT) has been observed. However, the factors driving these trends are yet to be fully understood. We examined the types of IDPs present in both mesophiles and thermophiles alongside those unique to just mesophiles. The shared group of IDPs exhibits similar disorder levels in the two groups of species, suggesting that certain IDPs unique to mesophiles may contribute to the observed decrease in IDP abundance as OGT increases. Subsequently, we used quasi-independent contrasts to explore the relationship between OGT and IDP abundance evolution. Interestingly, we found no significant relationship between OGT and IDP abundance contrasts, suggesting that the evolution of lower IDP abundance in thermophiles may not be solely linked to OGT. This study provides a foundation for future research into the intricate relationship between IDP evolution and environmental adaptation. Our findings support further research on the adaptive significance of intrinsic disorder in bacterial species.

## 1. Introduction

### 1.1. Intrinsically Disordered Proteins and Their Abundance

For a long time, a defined protein structure was thought to be essential for protein functionality; however, this notion has been challenged as the concept of intrinsically disordered proteins (IDPs) has been established. IDPs, sometimes referred to as inherently unstructured proteins or nonfolding proteins, are proteins that lack a stable structure. Renowned for their flexibility, IDPs can adopt diverse conformations, setting them apart from structured proteins. This structural dynamism allows IDPs to engage in a wide range of biochemical functions, underscoring their versatility in cellular regulation [1], signaling cascades, and intricate molecular interactions. Moreover, the structural disorder has been associated with an increase in both the number and variety of functions based on Swiss-Prot function tags [2].

The tendency towards intrinsic disorder in proteins can be predicted using protein amino acid (AA) composition. Disordered proteins have a higher proportion of hydrophilic and uncompensated charged AAs than ordered ones; therefore, physiochemical properties such as absolute mean charge and mean hydrophobicity can be used to classify proteins as ordered or disordered [3,4]. The charge-hydrophobicity phase space could be plotted, and such plots have been proven to be reliable predictors of protein disorder [3]. This concept is the core of the advanced computational tools predicting IDP/IDR, such as PONDR [4], IUPred [5], fIDPnn [6], and many more. These tools use protein AA composition and often incorporate a window-based analysis to predict intrinsic disorder in proteins. Specifically, disordered proteins tend to exhibit a higher proportion of hydrophilic and uncompensated charged AAs compared to ordered ones. The sliding window of AA along the sequence allows the assessment of local patterns and variations in physiochemical properties.

IDPs are widespread across all life domains [7] including viruses [8]. Their abundance is influenced by various factors, including, for example, organism complexity, with larger genomes generally displaying higher levels of IDPs [9]. Eukaryotes generally show both a higher frequency [10,11] and longer lengths of IDPs [12,13,14] compared to prokaryotes. Notably, within prokaryotes, IDP abundance is influenced by optimal growth temperatures (OGT) being significantly larger in mesophiles than in extremophiles adapted to higher temperatures [15,16]. These results challenge the conventional understanding of the advantageous role of IDPs in extreme conditions, as they play an important role in detecting changes in the environment [17].

Studying IDP abundance is essential for understanding cellular functioning, regulatory systems, and adaptive responses to the environment, as well as providing insights into their evolution across proteomes. IDPs play diverse roles in many cellular processes, and understanding factors influencing IDP abundance provides a key to unraveling the dynamic and flexible nature of these proteins, shedding light on their functional significance in cellular systems. While existing findings do describe a general pattern, specific causal elements underlying the association between OGT and IDP abundance remain unknown, offering an intriguing knowledge gap. The question at hand is determining if mesophiles have a larger number of IDPs or instead possess analogous proteins at greater disorder levels. In addition, the apparent connection between OGT and IDP abundance could be a phylogenetic consequence rather than a direct result of the OGT effect on IDP abundance. Our research strives to go beyond existing boundaries to answer these questions. To accomplish this, we assessed IDP abundance within distinct groups of IDPs. Additionally, we conducted a quasi-independent contrast calculation to gauge the impact of phylogeny on the relationship between OGT and IDP abundance.

### 1.2. Identification of IDP Groups

IDPs can be classified in many ways, including based on their molecular functions, functional features, sequence conservation, expression patterns, and biophysical properties [18]. As an extension of using AA physiochemical properties to calculate protein absolute mean charge and mean hydrophobicity, more complex parameters, such as the fraction of charged residues (FCR) and net charge per residue (NCPR), can be used to separate IDPs into strong polyelectrolytes, strong polyampholytes, boundary, and weak IDPs [19]:

Weak polyampholytes/polyelectrolytes (region 1): contain a small number of both positively and negatively charged AAs, as well as an approximately neutral overall charge. These proteins are often globules and tadpoles. 

Boundary proteins (region 2, also known as Janus sequences): proteins that resemble both region 1 and region 3 properties. Specific properties of these proteins are largely context-dependent. 

Strong polyampholytes (region 3): contain a significant number of both positively and negatively charged AAs, as well as an approximately neutral overall charge. These proteins are often flexible and form distinctly nonglobular coil-like, hairpin-like, or chimeric conformations. 

Negative strong polyelectrolytes (region 4) and positive strong polyelectrolytes (region 5): contain a large number of either positively or negatively charged AAs, which results in either a strongly positive or a strongly negative overall charge. These proteins are often very flexible and form swollen coil-like conformations.

This type of classification also allows the separation of globules from swollen coils [20]. In addition to the above classification, we grouped proteins based on their AA similarity and reported molecular functions where they were available. Finally, we identified clusters of similar proteins across thermophiles and mesophiles to detect any potential differences in disorder levels between the two species groups. We visualized aligned disorder values for the most divergent clusters in order to assess whether there are any patterns leading to that divergence.

### 1.3. Quasi-Independent Contrasts

Analysis of quasi-independent contrasts is an important method that helps us to unravel the relationships between OGT, IDP abundance, and phylogenetic relationships between the bacterial species used in the analysis. By employing quasi-independent contrasts, we can control for shared ancestry among species, ensuring a more accurate assessment of the direct impact of OGT on IDP abundance. The need for phylogeny-based comparative methods becomes evident when examining relationships between genes, phenotypes, and environmental factors among related species. Traditional statistical methods may be inadequate for quantifying these relationships due to the inherent co-ancestry among data points.

Independent and quasi-independent contrast comparison offers a more sophisticated means of addressing this challenge. As described by Xia [21], the method involves the minimization of the residual sum of squares by inferring ancestral states, accounting for phylogenetic influences through weighting factors. By applying quasi-independent contrasts, we can assess the relationship between OGT and IDP abundance while accounting for phylogenetic relationships, thus providing a more precise evaluation of how OGT directly influences IDP abundance. The contrasts between the two variables can be fitted into a linear model with an intercept fixed at the origin, and that model can then be interpreted to provide additional insights.

## 2. Results and Discussion

### 2.1. Overall IDP Abundance in Different Proteomes

We identified a weak negative relationship between OGT and FOD predicted by RAPID (Figure 1). This result partially supports previous findings that thermophiles should have a lower IDP abundance. The relationship is significant, but the effect size seems to be very small (R^2^ = 0.016, slope = −0.0003, and *p*-value = 0.030). Additionally, when separating species into thermophilic (OGT of at least 40 °C) and mesophilic (all other species) and comparing their FOD as predicted by RAPID (Figure 2) using a two-sided *t*-test, a significant difference was observed with mesophiles having more disorder (*p*-value = 6.898 × 10^−43^). However, the effect size is very small: the thermophilic average FOD = 0.1301 ± 0.0004 and the mesophilic average FOD = 0.1364 ± 0.0001. The high significance is very likely the result of the large sample size in this case and not the strength of the relationship.

The effect of OGT on FOD can also be seen in the decreased variation in FOD as ODT increases (Figure 1). With low OGT, FOD can be low or high. However, high OGT might seem to be selected against high FOD, pushing the variation in FOD to a lower range.

### 2.2. Overall IDP Abundance in Orthologs

For each cluster identified using CD-HIT as described in Materials and Methods, we recalculated disorder predictions using fIDPnn, a more accurate but also a much slower model. FOD calculations were found to be highly correlated between RAPID and fIDPnn, so the use of RAPID as a fast initial filter model has been justified (Appendix A).

The already weak negative relationship between OGT and FOD that we observed for the overall dataset (Figure 1) has not been seen for cluster data when using all clustered proteins as a subset. Surprisingly, an unexpected positive relationship emerges for both RAPID and fIDPnn-predicted FOD (Figure 3). This intriguing finding challenges the notion that orthologous proteins shared between thermophiles and mesophiles have greater disorder levels in mesophiles. Comparison of mean FOD values across the two datasets also produced opposite results from those of overall proteomes (Figure 4).

### 2.3. Abundance in Different IDP Classes and Proteins with Different Molecular Functions

Utilizing the classification based on the fraction of charged residues (FCR) and net charge per residue (NCPR) of amino acid sequences, clustered orthologs were categorized into five distinct classes: weak polyampholytes/polyelectrolytes, boundary proteins, strong polyampholytes, negative strong polyelectrolytes, and positive strong polyelectrolytes. Interestingly, none of the orthologs were classified as negative strong polyelectrolytes, while all other classes were represented by some clustered proteins. The corresponding fraction of disorder (FOD) values for each class, as predicted by the fIDPnn model, are presented in Table 1.

For each class, the table provides the mean FOD value along with its standard error, denoted as ±, and the sample size (n) representing the number of proteins within each category. Comparable levels of disorder between thermophiles and mesophiles were found for boundary proteins and strong polyampholytes, but some differences could be observed between the two species groups for weak polyampholytes/polyelectrolytes and positive strong polyelectrolytes. Thermophilic weak polyampholytes/polyelectrolytes had more disorder than mesophilic ones (FOD of 0.303 for thermophiles and 0.192 for mesophiles, *t*-test *p*-value = 0.004). Conversely, positive string polyelectrolytes were found to be more disordered in mesophiles than in thermophiles, although this effect was not found to be statistically significant (FOD of 0.682 for thermophiles and 0.706 for mesophiles, *t*-test *p*-value = 0.320). This finding may be a possible explanation for the negative correlation observed in Figure 1 and could be explained by the higher compactness of IDPs in higher temperatures [22].

Similarly to the above, we calculated the average FOD for identified molecular functions, as tagged on UniProt (Table 2). We found that IDP orthologs tagged as activator and nuclease were unique to only mesophiles, although they were not found at large levels there either–only 33 activators and 18 nucleases. At the same time, activator proteins had a relatively high FOD of 0.339, as predicted by fIDPnn. Moreover, we did not observe differences in FOD levels across any of the molecular functions that had been identified for orthologs in both mesophiles and thermophiles, especially among the more disordered ones.

The abovementioned results indicate that thermophiles are more likely to lack some IDPs that are present in mesophiles than to have less disordered orthologs in most cases. At the same time, a slight increase in the average FOD has been seen for mesophilic coil-like proteins, which are often involved in signaling through binding to various partners [23,24]. On the other hand, weak polyampholytes and polyelectrolytes might be more disordered in thermophiles because these IDPs may be more involved with adaptations to high temperatures. Interestingly, we were able to find examples of disorder differences between thermophilic and mesophilic weak polyampholytes/polyelectrolytes going in both directions. Large ribosomal subunit protein uL11 orthologs (UniProt IDs A0A7V5PNC3, A0A291PC16, A0A1B4VGG8, A0A250KZM7, and A0A5C1EBZ5, among others) were generally more disordered in thermophiles. Conversely, small acid-soluble spore protein sspB orthologs (UniProt IDs A0A0D8BNT0, A0A0D8BRQ7, A0A2K9J164, A0A0U4FDZ6, and A0A221MG13, among others) were found to be more disordered in mesophiles.

Among the IDPs from clusters that turned out to be unique to mesophiles, the majority were tagged as either ribosomal or rRNA-binding proteins (714 and 607 IDs out of 3469 proteins with tagged molecular functions). Additionally, some were tagged as transferase (278), chaperone (220), tRNA-binding (196), and DNA-binding (190). Apart from transferases and, partially, chaperones, all these groups are generally short proteins with significant disorder levels. We can also see that proteins with the same molecular functions are abundant in thermophiles, and it is possible that the large number of clusters being unique to mesophiles is due to the large number of variants of these proteins in general across all the species.

### 2.4. Analysis of Aligned Ortholog Clusters

Given the results of cluster analysis from the previous section, it seems that the relationship between OGT and IDP abundance is a complex one, and a look into the nature of the aligned clusters may reveal some patterns between IDP AAs and levels of disorder. We identified 10 ortholog clusters with the largest absolute differences between mesophilic and thermophilic FOD (Table 3). Among these, six had thermophilic IDPs that were more disordered than their mesophilic orthologs, and four were more disordered in mesophiles. The larger FOD within each cluster is underlined. The majority of these proteins turned out to be ribosomal proteins, although we also identified a rubredoxin, an acyl carrier, a spore protein, and a cupin protein among the clusters.

The identified clusters have been aligned and visualized in order to investigate any potential patterns and regions that contribute most to the observed differences in disorder levels (Appendix A). Additionally, WebLogo [25] diagrams have been created for these alignments for the assessment of AA consensus sequences. Hydrophobic and acidic AAs seem to be prevalent in regions where thermophilic IDP has a higher level of disorder, possibly indicating some temperature sensitivity of these residues. Conversely, polar AAs seem to be more frequent in IDPs that show larger disorder in mesophiles, although these AAs are generally common in IDPs. Combined with the other results, these findings suggest that neither the functional background of IDPs nor their AA composition have simple relationships with the levels of disorder in mesophiles and thermophiles. Instead, the relationship is a highly complex one, and further research into these factors’ contribution to IDP formation would be beneficial.

### 2.5. Phylogeny Impact on FOD/OGT Relationship

In our study, we tried to assess whether the weak negative correlation between OGT and FOD, as observed in the overall proteome comparison, persists when accounting for phylogeny using quasi-independent contrasts. Surprisingly, our findings indicate that phylogeny exerts a more substantial influence on the relationship than OGT in bacterial species. Contrary to the initial observation in the overall dataset, the weak negative relationship has not been observed for contrast data (R^2^ = 0.002, slope = 8.552 × 10^−5^, and *p*-value = 0.491). The scatter plot of the contrasts, illustrated in Figure 5, suggests an absence of any noticeable relationship, implying that OGT may not be a decisive factor influencing IDP abundance. Instead, it appears to be a characteristic carried along with the relative taxa, adding a nuanced layer to our understanding of factors affecting IDP abundance in bacteria.

Several possible factors may be driving the observed phylogenetic effects. First, variables related to genomic characteristics may contribute to the observed differences in IDP abundance. For example, genome size and GC content are known to be correlated with IDP abundance [26], and other genomic features associated with phylogeny could also influence the evolution of IDPs independent of OGT. Second, the co-evolution of protein networks within specific phylogenetic groups may play a role in shaping IDP evolution. Interactions between proteins and their partners, influenced by shared evolutionary history, could contribute to the observed patterns in disorder abundance. Lastly, shared ancestry and evolutionary relationships may contribute to the observed patterns in IDP abundance, with closely related bacterial species inheriting similar traits, including features related to IDPs, such as amino acid composition, structural motifs, or functional roles in cellular processes. These traits may be unrelated to the OGT of the species but have an effect on IDP abundance.

These conclusions are particularly intriguing since they add context to previous findings showing thermophiles had lower disorder abundance than mesophiles. The lack of a clear functional justification for these findings suggests that factors other than OGT should be considered. Our findings call for a reconsideration of the relationship between IDP abundance and environmental conditions, emphasizing the importance of phylogeny and potentially other variables such as genome size. Future research should take into account this complex network of elements in order to improve the accuracy and comprehensiveness of investigations in unraveling the complexity of IDP evolution in bacterial species.

## 3. Materials and Methods

### 3.1. Data Sources and Availability

We used UniProt [27] as a source for the AA sequence data and TEMPURA [28] for bacteria OGT data. TEMPURA contains bacteria and archaea OGT with ribosomal 16s gene used for reference. We downloaded the database entirely and then filtered it to only contain data for bacteria with recorded 16s accession numbers.

For each of the remaining species, we searched for a reference proteome on UniProt and downloaded them if they were available and had at least 1000 proteins. As a result, our dataset consisted of 1,132,382 proteins from 304 species.

### 3.2. Protein Clustering

All proteins in our dataset have been clustered using CD-HIT [29] with a setting for a minimal global similarity score of 0.7. Clusters have been filtered to follow these conditions:Proteins from at least 10 different species per cluster;At least one candidate IDP (identification described in the disorder calculation subsection);

This way, we generated 616 clusters of interest. Additionally, we obtained UniProt molecular function tags for each protein from these clusters.

### 3.3. Disorder Calculations

Two rounds of disorder prediction have been performed:

First, we calculated disorders for each protein of the dataset using RAPID [30]. While this model has the disadvantage of only predicting a single overall disorder metric for a given protein, it is in, fact, rapid and, given the large number of proteins in our dataset, is a suitable model for initial filtering. Additionally, we calculated the FCR and NCPR for each protein to group them into five classes, as described by Das and Pappu [19].

Based on the RAPID results and FCR/NCPR, we identified IDP candidates as those that satisfy at least one of the following conditions:3.RAPID disorder score ≥ 0.5;4.Total number of residues × RAPID disorder score ≥ 100;5.IDP type 3, 4, or 5 (strong polyampholytes or positive/negative strong polyelectrolytes);

This way, we significantly narrowed down the list of proteins for further analysis as well as defined a binary ordered/disordered separation for our dataset.

The second round of disorder calculation was performed on clusters of interest (see the relevant section about clustering for more information) using fIDPnn [6], a more advanced but also much more time-consuming method than RAPID. This model has shown to be a very effective one [31,32], as well as able to output a disorder score for each residue of the protein, which allowed us to compare IDPs at the residue level. By using the two-round approach, we were able to evaluate entire proteomes using a faster model, identify potential IDPs, and then obtain more detailed results for these candidates using a more complex but slower model.

### 3.4. Cluster Disorder Alignment

The AA sequences of each protein in a cluster were aligned using ClustalW [33] with a gap insertion penalty of 1 and a gap extension penalty of 0.5. The disorder scores were smoothed for the plots with a moving average. The sliding window was equal to protein length divided by 30.

### 3.5. Quasi-Independent Contrast Calculation

We referred to Xia’s least squares method of quasi-independent contrast calculation [21]. First, we obtained 16s nucleotide sequences based on TEMPURA accession numbers for all bacteria, with an addition of one sequence from archaea species, NG_046384.1 of *Pyrobaculum ferrireducens*, which was used as an outgroup. Four of the sequences were removed using DAMBE [34], as they were for entire genomes rather than 16 s. We aligned the resulting sequences using MAFFT [35] with default parameters except for specification for nucleotide sequences and calculated the distances based on the aligned sequence identity. The distances then have been used to build a phylogenetic tree using UPGMA and NG_046384.1 as outgroups.

For each leaf of the resulting tree, which represented one of the species from our dataset, we collected an overall fraction of disorder (FOD), as predicted by RAPID and OGT, as recorded in TEMPURA. For each internal node, average FOD and OGT were used as initial guesses and were later optimized using RSS minimization. Finally, contrasts have been calculated between each offspring pair sharing the same ancestor.

## Figures and Tables

**Figure 1 ijms-25-02000-f001:**
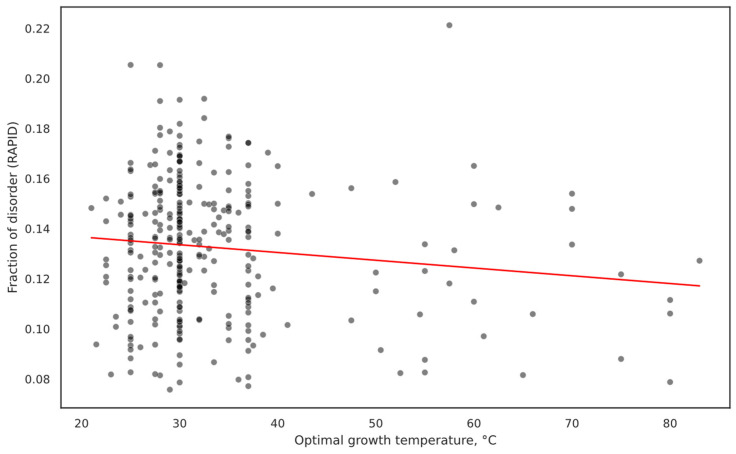
Scatter plot of OGT and FOD, with line showing OLS model. High OGT is associated with lower FOD. Effect size seems to be small but statistically significant (R^2^ = 0.016, slope = −0.0003, and *p*-value = 0.030) for the linear model. Majority of organisms are mesophiles, which could potentially skew the results of modeling.

**Figure 2 ijms-25-02000-f002:**
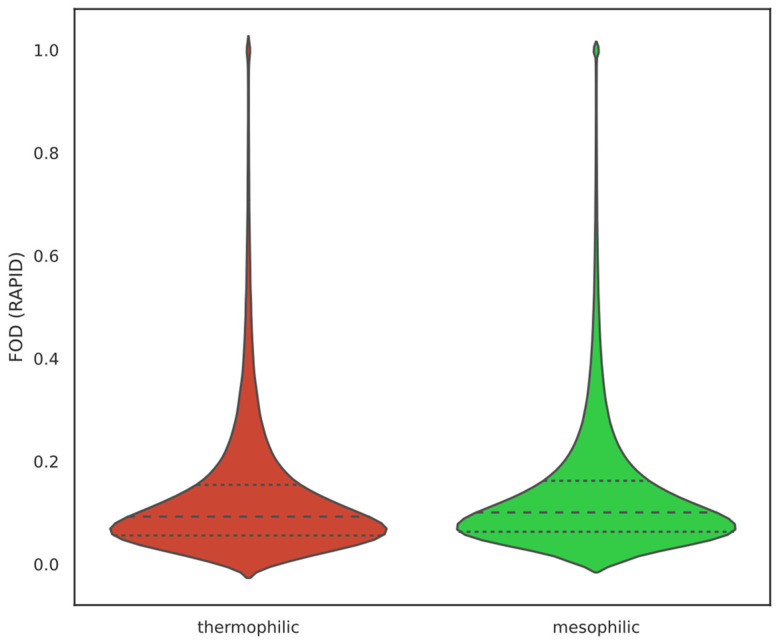
FOD distributions for thermophilic and mesophilic proteins. The two distributions seem to be very similar even though a statistically significant difference has been observed between the mean values (*t*-test *p*-value = 6.898 × 10^−43^). This significance is likely to be the result of the large sample size. Thermophilic average FOD = 0.1301 ± 0.0004 and mesophilic average FOD = 0.1364 ± 0.0001. All proteins from the dataset have been assessed, and FOD has been predicted using RAPID.

**Figure 3 ijms-25-02000-f003:**
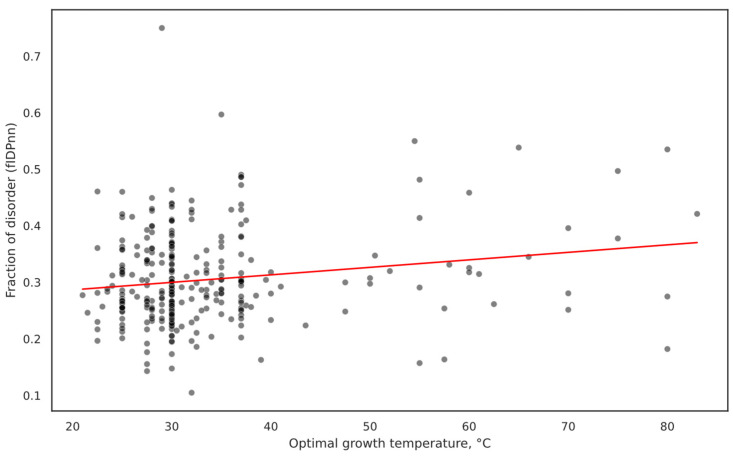
Scatter plot of OGT and FOD, with line showing OLS model. Positive relationship between OGT and FOD is observed for clustered data. R^2^ = 0.052, slope = 0.0017, and *p*-value = 5.69 × 10^−5^ for the linear model. FOD has been calculated using fIDPnn. Linear model with RAPID FOD showed similar results (R^2^ = 0.031, slope = 0.0013, and *p*-value = 0.002). The positive relationship is opposite to the one for overall data (Figure 1).

**Figure 4 ijms-25-02000-f004:**
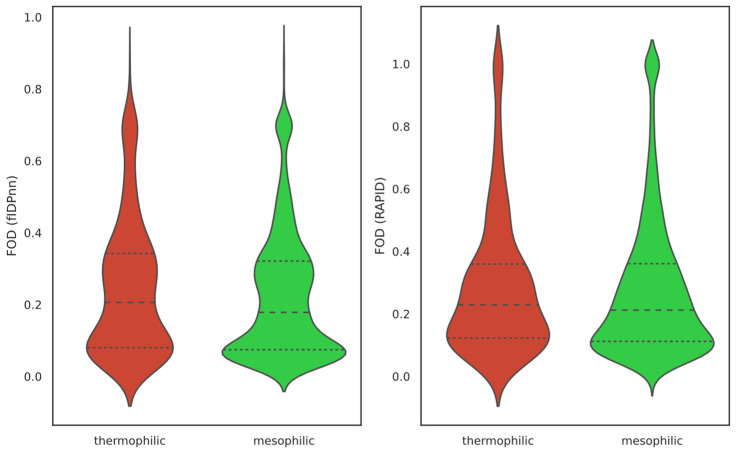
FOD distributions for thermophilic and mesophilic orthologs. Left violin plot shows distributions of FOD calculated by fIDPnn, and right violin plot shows distributions of FOD calculated by RAPID. The pairs of distributions seem to be very similar even though a statistically significant difference has been observed between the mean values of fIDPnn FOD (*t*-test *p*-value = 0.0025 for fIDPnn and 0.167 for RAPID). Using fIDPnn, thermophilic average FOD = 0.2425 ± 0.007 and mesophilic average FOD = 0.2232 ± 0.002. Using RAPID, thermophilic average FOD = 0.2887 ± 0.009 and mesophilic average FOD = 0.2760 ± 0.003. Interestingly, the differences are in opposite directions from the overall data (Figure 2).

**Figure 5 ijms-25-02000-f005:**
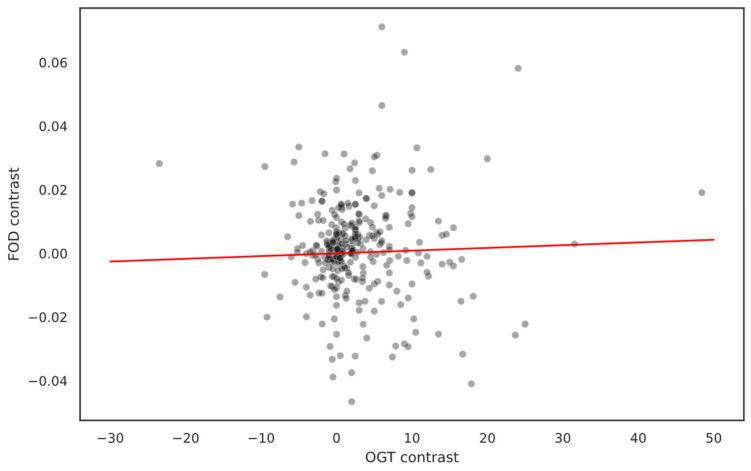
Scatter plot of FOD contrast/OGT contrast relationship. Contrast controls for phylogeny impact, and we observe no relationship between IDP abundance and temperature differences; therefore, phylogeny seems to be a more important factor than OGT.

**Table 1 ijms-25-02000-t001:** FOD for different classes of ortholog IDPs in mesophilic and thermophilic bacteria.

IDP Class	Thermophilic FOD	Mesophilic FOD
Weak polyampholytes/polyelectrolytes	0.303 ± 0.054; *n* = 21	0.192 ± 0.007; *n* = 609
Boundary proteins	0.181 ± 0.007; *n* = 422	0.176 ± 0.002; *n* = 7830
Strong polyampholytes	0.358 ± 0.014; *n* = 180	0.357 ± 0.004; *n* = 2145
Negative strong polyelectrolytes	-	-
Positive strong polyelectrolytes	0.682 ± 0.016; *n* = 13	0.706 ± 0.005; *n* = 256

**Table 2 ijms-25-02000-t002:** FOD for different function tags of ortholog IDPs in mesophilic and thermophilic bacteria.

IDP Function Tag	Thermophilic FOD	Mesophilic FOD
Activator	-	0.339 ± 0.010; *n* = 33
Nuclease	-	0.110 ± 0.003; *n* = 18
Chaperone	0.090 ± 0.010; *n* = 18	0.114 ± 0.003; *n* = 516
DNA-binding	0.276 ± 0.032; *n* = 35	0.264 ± 0.008; *n* = 477
Elongation factor	0.125 ± 0.034; *n* = 33	0.070 ± 0.005; *n* = 367
Excision nuclease	0.053 ± 0.005; *n* = 7	0.054 ± 0.001; *n* = 192
Hydrolase	0.064 ± 0.005; *n* = 14	0.108 ± 0.009; *n* = 209
Initiation factor	0.217 ± 0.026; *n* = 3	0.209 ± 0.004; *n* = 56
Isomerase	0.090 ± 0.024; *n* = 3	0.076 ± 0.001; *n* = 91
Ligase	0.060 ± 0.008; *n* = 12	0.058 ± 0.002; *n* = 336
Lyase	0.077 ± 0.005; *n* = 11	0.076 ± 0.001; *n* = 230
Multifunctional enzyme	0.055 ± 0.000; *n* = 1	0.052 ± 0.001; *n* = 13
Oxidoreductase	0.102 ± 0.018; *n* = 13	0.073 ± 0.002; *n* = 245
Peroxidase	0.082 ± 0.007; *n* = 3	0.097 ± 0.004; *n* = 59
Protease	0.082 ± 0.008; *n* = 7	0.080 ± 0.001; *n* = 284
RNA-binding	0.165 ± 0.033; *n* = 17	0.126 ± 0.005; *n* = 388
Receptor	0.095 ± 0.000; *n* = 1	0.112 ± 0.016; *n* = 10
Repressor	0.331 ± 0.086; *n* = 4	0.263 ± 0.017; *n* = 51
Ribosomal protein	0.467 ± 0.018; *n* = 110	0.437 ± 0.004; *n* = 1859
Rotamase	0.191 ± 0.030; *n* = 4	0.209 ± 0.010; *n* = 36
Serine protease	0.062 ± 0.000; *n* = 1	0.063 ± 0.001; *n* = 49
Sigma factor	0.142 ± 0.011; *n* = 18	0.149 ± 0.003; *n* = 154
Topoisomerase	0.075 ± 0.007; *n* = 6	0.087 ± 0.001; *n* = 151
Transferase	0.087 ± 0.012; *n* = 27	0.073 ± 0.003; *n* = 713
Translocase	0.049 ± 0.004; *n* = 7	0.076 ± 0.004; *n* = 127
rRNA-binding	0.294 ± 0.010; *n* = 96	0.284 ± 0.002; *n* = 1700
tRNA-binding	0.338 ± 0.015; *n* = 55	0.297 ± 0.005; *n* = 686

**Table 3 ijms-25-02000-t003:** Most divergent FOD between thermophilic and mesophilic orthologs.

Protein Name	FOD (Thermophilic)	FOD (Mesophilic)	Absolute FOD Difference	Cluster Members
Rubredoxin	0.085	0.389	0.304	A0A291P6P4, A0A410H536, A0A1B2LXP3…
Acyl carrier	0.534	0.392	0.142	A0A291P5S3, A0A410H1W4, A0A386X534…
Spore protein	0.647	0.786	0.139	A0A0D8BNT0, A0A0D8BRQ7, M5R4X2…
LRSP * bL19	0.357	0.239	0.118	A0A1U9K6D3, A0A1B9NF78, A0A1B0ZK26…
SRSP bS21	0.596	0.484	0.112	A0A0D5YVA4, A0A1Z4BT12, A0A1L3J4J5…
SRSP uS14	0.476	0.370	0.106	A0A0P0DDQ1, A0A0D5YRD0, A0A0S2I2L9…
LRSP bL28	0.512	0.610	0.098	A0A0D8BU85, M5QWZ7, A0A1D7QW46…
Cupin	0.385	0.287	0.097	A0A0K2SHK7, A0A0D5NPB9, A0A4P6K4Z4…
LRSP uL24	0.323	0.414	0.090	A0A0D8BQ30, M5QVZ2, A0A1D7QZW9…
SRSP uS14	0.501	0.417	0.083	A0A291PBX7, A0A7C9NQP7, A0A3T1DHB4…

* LRSP = large ribosomal subunit protein; SRSP = small ribosomal subunit protein.

## Data Availability

All data files and code scripts are available at https://github.com/alibekk93/IDP_analysis (accessed on 4 February 2024).

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
