# Peer review of "Mesophiles vs. Thermophiles: Untangling the Hot Mess of Intrinsically Disordered Proteins and Growth Temperature of Bacteria"

_ijms, 2024, doi:10.3390/ijms25042000_

Round 1
Reviewer 1 Report
Comments and Suggestions for Authors
1) Lack of Functional Justification: While the study provides a comprehensive analysis of IDP abundance and characteristics, it lacks a detailed exploration of the functional implications of these findings. The authors mention that certain IDPs unique to mesophiles may contribute to the observed decrease in IDP abundance with increasing OGT, but they do not delve into specific examples or mechanisms to support this hypothesis.
2) Limited Discussion of Phylogenetic Impact: The results of the quasi-independent contrasts analysis suggest that phylogeny may have a more substantial influence on IDP abundance than OGT. However, the authors do not thoroughly discuss the potential factors driving this phylogenetic effect or explore the implications of these findings for understanding IDP evolution.
3) Insufficient Exploration of IDP Clusters: The identification of clusters of similar proteins across thermophiles and mesophiles presents an opportunity for in-depth comparative analyses. However, the authors have not fully exploited this aspect of the study. A more detailed examination of the aligned disorder values for the most divergent clusters could provide insights into specific regions or motifs that contribute to the observed differences in disorder levels.
Author Response
Dear Reviewer,
We wish to thank you for you time and effort dedicated to generating reviews. Your thoughtful comments and suggestions have resulted in significant improvement of our manuscript. In the current version we have revised the manuscript to address your comments and to improve clarity and flow. Our replies are in bold italic and the changes in the manuscript file have been highlighted with yellow.
1) Lack of Functional Justification: While the study provides a comprehensive analysis of IDP abundance and characteristics, it lacks a detailed exploration of the functional implications of these findings. The authors mention that certain IDPs unique to mesophiles may contribute to the observed decrease in IDP abundance with increasing OGT, but they do not delve into specific examples or mechanisms to support this hypothesis.
Thank you for the comment and suggestions. We expanded subsection 2.3 to add some specific examples of clusters with different observed differences between thermophilic and mesophilic orthologs. Additionally, we describe the results for clusters of IDP which only contained mesophilic proteins. We hope that the additional information will aid the discussion of our results and the conclusion of complexity of IDP / OGT relationship.
2) Limited Discussion of Phylogenetic Impact: The results of the quasi-independent contrasts analysis suggest that phylogeny may have a more substantial influence on IDP abundance than OGT. However, the authors do not thoroughly discuss the potential factors driving this phylogenetic effect or explore the implications of these findings for understanding IDP evolution.
Many thanks for the suggestion. We expanded subsection 2.5 to discuss the results more thoroughly. Specifically, we highlight three potential factors driving the observed phylogenetic effect: 1) genomic features, such as genome size and GC content; 2) protein networks co-evolution impact; 3) inheritance of features unrelated to OGT, but with effect on IDP abundance. We hope that the revision will improve discussion in the manuscript.
3) Insufficient Exploration of IDP Clusters: The identification of clusters of similar proteins across thermophiles and mesophiles presents an opportunity for in-depth comparative analyses. However, the authors have not fully exploited this aspect of the study. A more detailed examination of the aligned disorder values for the most divergent clusters could provide insights into specific regions or motifs that contribute to the observed differences in disorder levels.
This is a very interesting point and we thank you for the suggestion. Indeed, there is potential in analyzing AA sequences of aligned clusters in order to find specific motifs that are associated with differences in disorder between mesophiles and thermophiles in these clusters. We believe that this analysis can be very promising and a separate extensive project should be performed to study all the clusters, not only the most diverse ones. We added WebLogo diagrams and discussed the aligned clusters results in a new subsection to highlight that potential in this manuscript.
Thank you very much for your help with improving the manuscript.
Reviewer 2 Report
Comments and Suggestions for Authors
The manuscript entitled “Mesophiles vs. Thermophiles: Untangling the Hot Mess of In-trinsically Disordered Proteins and Growth Temperature of Bacteria” has many mistakes, authors need to rectify many portions.
o Why has the notion of a defined protein structure being essential for protein functionality been challenged, and what is the significance of intrinsically disordered proteins (IDPs) in cellular regulation?
o How can the tendency towards intrinsic disorder in proteins be predicted using protein amino acid composition, and what physiochemical properties are considered in classifying proteins as ordered or disordered?
o What factors influence the abundance of IDPs, and how does their prevalence vary among different life domains, especially between eukaryotes and prokaryotes?
o Why is studying IDP abundance crucial for understanding cellular functioning, regulatory systems, and adaptive responses to the environment?
o How does the research aim to go beyond existing boundaries in understanding the relationship between optimal growth temperatures (OGT) and IDP abundance?
o How can IDPs be classified based on functional features, sequence conservation, expression patterns, and biophysical properties?
o What are the different regions of IDPs based on metrics such as fraction of charged residues (FCR) and net charge per residue (NCPR), and what are the characteristics of proteins in each region?
o How does the classification of IDPs into different regions help in understanding their structural properties, such as globules, tadpoles, and coil-like conformations?
o Why is the analysis of quasi-independent contrasts important in unraveling the relationships between OGT, IDP abundance, and phylogenetic relationships among bacterial species?
o How does the quasi-independent contrast method control for shared ancestry among species and enable a more accurate assessment of the direct impact of OGT on IDP abundance?
o What relationship between optimal growth temperatures (OGT) and Fraction of Disorder (FOD) was identified, and how does it compare to previous findings regarding thermophiles and their IDP abundance?
o Why is the weak negative relationship between OGT and FOD, as predicted by RAPID, considered significant, and what is the effect size of this relationship?
o When species are separated into thermophilic and mesophilic categories, what statistically significant difference in FOD is observed, and how does it relate to the large sample size in this case?
o In Figure 1, how is the majority of organisms being mesophiles potentially influencing the observed relationship between OGT and FOD?
o How does the effect of OGT on FOD manifest in the variation of FOD as OGT increases, and what implications does this have for understanding IDP abundance?
o Why was RAPID chosen as the fast initial filter model, and how does it compare to fIDPnn in terms of disorder predictions for orthologs?
o What unexpected relationship between OGT and FOD is observed when considering cluster data for orthologs, and how does it differ from the overall dataset?
o In Figure 3, what is the significance of the positive relationship between OGT and FOD for clustered data, and how does it contrast with the overall proteome results?
o How do the FOD distributions for thermophilic and mesophilic orthologs compare in Figure 4, and what does this reveal about the disorder levels in orthologous proteins between the two categories?
o How are clustered orthologs categorized based on the Fraction of Charged Residues (FCR) and Net Charge Per Residue (NCPR) of amino acid sequences, and what are the observed differences in disorder levels between thermophiles and mesophiles for each category?
Good Luck!
Comments on the Quality of English LanguageModerate modification required
Author Response
Dear Reviewer,
We wish to thank you for you time and effort dedicated to producing the reviews. Your thoughtful comments and suggestions have resulted in significant improvement of our manuscript. In the current version we have revised the manuscript to address your comments and to improve clarity and flow.
The changes in the file have been highlighted with yellow in the updated file, here is the summary of changes for your convenience:
- introduction of IDPs and their role in challenging the conventional notion of a defined protein structure as essential for functionality has been edited to improve clarity;
- more detailed description of how advanced disorder prediction tools use AA composition;
- revised description of factors that affect abundance of IDPs in different organisms to improve readability and add context;
- clarified the goals and importance of the study for readers' better understanding;
- IDP classification options have been described more thoroughly;
- revision of introduction to quasi-independent contrasts to improve readability and clarify the need for quasi-independent analysis in studying OGT / IDP abundance relationship;
- text in subsection 2.1 and caption under Figure 1 have been updated for better clarity;
- revision of subsection 2.2 and caption under figures 3 and 4 to improve readability;
- more clarification was added to explain the motivation behind using RAPID as the first model;
Additionally, here are our replies to your comments point-by-point, highlighted in bold and italic:
o Why has the notion of a defined protein structure being essential for protein functionality been challenged, and what is the significance of intrinsically disordered proteins (IDPs) in cellular regulation?
Historically, it has been thought that a stable protein structure was absolutely essential for the protein to be functional, however this notion had been challenged as the significance of IDPs has been recognized. Unlike structured proteins, IDPs lack a stable, predetermined structure, yet play crucial roles in many biological processes, including cellular regulation.
o How can the tendency towards intrinsic disorder in proteins be predicted using protein amino acid composition, and what physiochemical properties are considered in classifying proteins as ordered or disordered?
Protein amino acid composition is the most important feature in disorder prediction, as physiochemical properties of the amino acids (such as the absolute mean charge and mean hydrophobicity) are used to classify proteins as ordered or disordered. Additionally, advanced disorder prediction tools use sliding windows to incorporate sectional information on AA properties.
o What factors influence the abundance of IDPs, and how does their prevalence vary among different life domains, especially between eukaryotes and prokaryotes?
IDP abundance in different species has been observed to be correlated with a number of factors, including organism complexity, GC content and optimal environments. Moreover, eucaryotes generally have more disorder than prokaryotes. While general patterns are seen, the mechanisms behind the observations are not always understood well. In our study we attempt to focus on prokaryotes and their optimal growth temperature to obtain more information on what affects IDP abundance in these species.
o Why is studying IDP abundance crucial for understanding cellular functioning, regulatory systems, and adaptive responses to the environment?
We believe that studying IDP abundance is very important in understanding cellular functioning, regulatory systems, and adaptive responses to the environment. IDPs play a large role in many of these processes, and more information on factors that affect IDP abundance will help in understanding their role in these processes. Moreover, prevalence of IDPs across different organisms can provide insights into the evolutionary aspects of these organisms and their cellular systems. Understanding IDP abundance can contribute to unraveling both the mechanisms of their activity as well as evolutionary process.
o How does the research aim to go beyond existing boundaries in understanding the relationship between optimal growth temperatures (OGT) and IDP abundance?
Our attempt is to add additional context to the knowledge base around IDP abundance / OGT relationship in prokaryotes. First, we predict disorder using entire proteomes of a large variety of species, rather than using IDP databases. The databases, while very convenient, often are limited in terms of the amount of data. Second, we cluster candidate IDPs in order to compare orthologous proteins. In previous studies, either specific proteins or known IDPs had been studied. Finally, we perform quasi-independent contrast analysis to control for phylogenetic features. This allows us to compare the groups of species in a more "fair" way as phylogenetic signal can distort the actual IDP abundance / OGT relationship.
o How can IDPs be classified based on functional features, sequence conservation, expression patterns, and biophysical properties?
Classification of IDPs can be performed in many ways as we specify some of them in the manuscript for context. Specifically, biophysical properties of AA sequences can be used to classify IDPs into 5 groups:
Weak polyampholytes/polyelectrolytes (region 1): contain a small number of both positively and negatively charged AA, as well as an approximately neutral overall charge. These proteins are often globules and tadpoles.
Boundary proteins (region 2, also known as Janus sequences): proteins that resemble both region 1 and region 3 properties. Specific properties of these proteins are largely context dependent.
Strong polyampholytes (region 3): contain a significant number of both positively and negatively charged AA, as well as an approximately neutral overall charge. These proteins are often flexible and form distinctly nonglobular coil-like, hairpin-like, or chimeric conformations.
Negative strong polyelectrolytes (region 4) and positive strong polyelectrolytes (region 5): contain a large number of either positively or negatively charged AA, which results in either a strongly positive or a strongly negative overall charge. These proteins are often very flexible and form swollen coil-like conformations.
o What are the different regions of IDPs based on metrics such as fraction of charged residues (FCR) and net charge per residue (NCPR), and what are the characteristics of proteins in each region?
The regions that we refer to in the manuscript are for FCR / NCPR plot and are described in the answer to the previous question. These regions, or categories of IDP, are based on protein absolute mean charge and mean hydrophobicity and are a common way of classifying proteins into more or less disordered.
o How does the classification of IDPs into different regions help in understanding their structural properties, such as globules, tadpoles, and coil-like conformations?
Negative strong polyelectrolytes (region 4) and positive strong polyelectrolytes (region 5) are known to be very flexible and form swollen coil-like conformations. Conversely, weak polyampholytes/polyelectrolytes (region 1) often form globules and tadpoles. This is based on the AA composition of the proteins.
o Why is the analysis of quasi-independent contrasts important in unraveling the relationships between OGT, IDP abundance, and phylogenetic relationships among bacterial species?
This is a very interesting question. Quasi-independent contrasts allow us to control for phylogenetic signal, which is a very strong factor in IDP abundance. Inherited factors may not be related to OGT, but influence IDP abundance. By controlling the phylogeny, we are able to add important context to the results that do not control for phylogeny.
o How does the quasi-independent contrast method control for shared ancestry among species and enable a more accurate assessment of the direct impact of OGT on IDP abundance?
Quasi-independent contrast method uses phylogenetic distances between the species as well as any other quantifiable features (IDP abundance and OGT in our case) to optimize contrast values between related species. This is done using minimization of residual sum of squares.
o What relationship between optimal growth temperatures (OGT) and Fraction of Disorder (FOD) was identified, and how does it compare to previous findings regarding thermophiles and their IDP abundance?
Previous findings typically reported that mesophiles have a larger abundance of IDP than that of thermophiles (as well as other extremophiles). We do see the same results for the overall data, although the effect size that we observe is very weak. At the same time, when using clusters of orthologs, we see a reversed relationship. Additionally, when using quasi-independent contrast method, we see no relationship. Our results are intriguing and suggest that future research into the intricate relationship between IDP evolution and environmental adaptation should also control for phylogenetics and work on ortholog data.
o Why is the weak negative relationship between OGT and FOD, as predicted by RAPID, considered significant, and what is the effect size of this relationship?
The significance of the relationship is statistical, as we find a very small p-value of 0.030. While the statistical significance seems to be present, we should be careful with any conclusions on this particular result as the effect size is very small (slope = -0.0003), which means that biologically the relationship is not likely to be significant. Furthermore, the other results that we report are converse to these ones.
o When species are separated into thermophilic and mesophilic categories, what statistically significant difference in FOD is observed, and how does it relate to the large sample size in this case?
For overall data, we observe that mesophiles have a slightly larger FOD than thermophiles. For clustered data, we see the opposite - thermophiles have a larger mean FOD than mesophiles. Sample sizes are quite large in both cases, but the first comparison has a much larger number of proteins. In both cases the effect sizes do not seem to be radical, but the power is large because of the sample sizes. Because of that, we need to be cautious with the interpretation of results.
o In Figure 1, how is the majority of organisms being mesophiles potentially influencing the observed relationship between OGT and FOD?
The number of mesophiles is much larger than that of thermophiles, which can be observed on Figure 1, as you can see more points on the left side of the scatter plot (indicating lower OGT) than on the right side of the plot (larger OGT). Both species groups have significant numbers of data, however, as the number of mesophiles is much larger, there may be some skewedness of the results. For the linear model, this should not be a problem, however we felt that this was an important thing to mention in the manuscript.
o How does the effect of OGT on FOD manifest in the variation of FOD as OGT increases, and what implications does this have for understanding IDP abundance?
For overall data, OGT and FOD generally show a weak negative correlation, indicating that as OGT increases, FOD tends to decrease. However, for clustered data, we see the opposite. This variation challenges conventional expectations, suggesting that the relationship between OGT and IDP abundance may be more nuanced than previously thought. We hope that our results provide additional insights into the factors influencing IDP abundance in prokaryote with different OGTs.
o Why was RAPID chosen as the fast initial filter model, and how does it compare to fIDPnn in terms of disorder predictions for orthologs?
RAPID was chosen as a model to predict disorder for the first round of prediction. This involved entire proteomes of all of our species and we needed a very fast model to be able to do it. RAPID is, as the name suggests, very fast at predicting protein disorder levels and, as our results show, is also an adequate model, as its results are well-correlated with the results from fIDPnn. For the second round of prediction, we only chose IDP candidates, so we were able to use fIDPnn for that. Using this approach, we were able to evaluate entire proteomes using a faster model, identify potential IDPs and then obtain more detailed results for these candidates using a more complex, but slower model.
o What unexpected relationship between OGT and FOD is observed when considering cluster data for orthologs, and how does it differ from the overall dataset?
For cluster dataset, we observe a weak positive correlation between OGT and FOD. For overall dataset, we see the opposite - a weak negative correlation between the two factors. The later is consistent with previous findings from similar experiments, however, the cluster analysis that we performed provide important context for our understanding of IDP / OGT relationship.
o In Figure 3, what is the significance of the positive relationship between OGT and FOD for clustered data, and how does it contrast with the overall proteome results?
The relationship on Figure 3 is the one for clustered dataset. While being weak (R2 = 0.052, slope = 0.0017), this relationship is statistically significant (p-value = 5.69e-05). It is opposite from the overall proteome results, which showed a weak negative correlation and provides important context for our understanding of IDP / OGT relationship.
o How do the FOD distributions for thermophilic and mesophilic orthologs compare in Figure 4, and what does this reveal about the disorder levels in orthologous proteins between the two categories?
FOD distributions on Figure 4 show IDP abundance for thermophilic and mesophilic clustered data. We can see that the abundance levels are very similar, although thermophiles have a slightly larger mean value. This is opposite of the overall data that we observed and adds important context for our understanding of IDP / OGT relationship.
o How are clustered orthologs categorized based on the Fraction of Charged Residues (FCR) and Net Charge Per Residue (NCPR) of amino acid sequences, and what are the observed differences in disorder levels between thermophiles and mesophiles for each category?
We use FCR and NCPR to categorize IDP into the 5 groups (listed and described in one of the previous answers). Comparable levels of disorder between thermophiles and mesophiles were found for boundary proteins and strong polyampholytes, but some differences could be observed between the two species groups for weak polyampholytes/polyelectrolytes and positive strong polyelectrolytes. Thermophilic weak polyampholytes/polyelectrolytes had more disorder than mesophilic ones. Conversely, positive string polyelectrolytes were found to be more disordered in mesophiles than in thermophiles, although this effect was not found to be statistically significant.
We hope that we were able to fix the errors and rectify the manuscript. Thank you again for your help.
Round 2
Reviewer 1 Report
Comments and Suggestions for Authors
The authors have satisfactorily addressed my concerns from the initial review. The additions and changes have improved the clarity and flow of the paper.